# Metal-Based Anticancer Complexes and p53: How Much Do We Know?

**DOI:** 10.3390/cancers15102834

**Published:** 2023-05-19

**Authors:** Samah Mutasim Alfadul, Egor M. Matnurov, Alexander E. Varakutin, Maria V. Babak

**Affiliations:** Drug Discovery Lab, Department of Chemistry, City University of Hong Kong, 83 Tat Chee Avenue, Hong Kong SAR 999077, China

**Keywords:** p53 family, zinc, copper, iron, ruthenium, platinum, metal anticancer complexes, bioinorganic

## Abstract

**Simple Summary:**

It is believed that metal complexes might be interesting alternatives to the small organic molecules for the treatment of cancer. Due to the variety of metal oxidation states and geometries, the structure of metal complexes can be easily modified based on the required design. For example, metal complexes can be specifically designed to interact with the p53 protein or its binding partners. The aim of this article is to discuss whether metal complexes can have a future as p53-targeting drugs.

**Abstract:**

P53 plays a key role in protecting the human genome from DNA-related mutations; however, it is one of the most frequently mutated genes in cancer. The P53 family members p63 and p73 were also shown to play important roles in cancer development and progression. Currently, there are various organic molecules from different structural classes of compounds that could reactivate the function of wild-type p53, degrade or inhibit mutant p53, etc. It was shown that: (1) the function of the wild-type p53 protein was dependent on the presence of Zn atoms, and (2) Zn supplementation restored the altered conformation of the mutant p53 protein. This prompted us to question whether the dependence of p53 on Zn and other metals might be used as a cancer vulnerability. This review article focuses on the role of different metals in the structure and function of p53, as well as discusses the effects of metal complexes based on Zn, Cu, Fe, Ru, Au, Ag, Pd, Pt, Ir, V, Mo, Bi and Sn on the p53 protein and p53-associated signaling.

## 1. P53 Family and Cancer

P53 is a tumor suppressor protein that plays a critical role in preventing tumor formation [1]. This protein is called the “guardian of the genome” due to its key role in protecting the genome from DNA damage-related mutations [2]. It is known that p53 acts as a transcriptional factor that detects DNA damage and other cellular stresses such as hypoxia, metabolic alternation, etc., and induces the appropriate response, such as cell cycle arrest, DNA repair, or apoptosis [3,4]. In healthy individuals, cells are characterized by low steady levels of the p53 protein due to its short half-life time (e.g., 6 min in the spleen of a normal adult mouse) [5]. Moreover, most of the time p53 exists in a largely inactive state, and its inactivation occurs through the binding with the E3 ubiquitin-protein ligase MDM2 protein [6]. In the case of DNA damage, cells stimulate the production of specific kinases which phosphorylate p53 at the MDM2 binding site [7]. As a result, the activated form of the p53 protein cannot be inhibited by MDM2, and gains the ability to bind to certain regions of the DNA, called p53-response elements [8]. This process results in the transcriptional activation of pro-apoptotic genes and the initiation of programmed cell death. Therefore, one of the main functions of the p53 protein is the removal of potentially oncogenic cells from the pool of replicating cells [3,9]. 

The *TP53* gene, which encodes for the p53 protein, is mutated in the majority of human cancers [10,11]. The most commonly occurring mutations of p53 include loss of function or missense mutations in the DNA binding domain [12,13]. In some of the p53-mutated cancer cells, the interaction between p53 and MDM2 is lost, leading to high levels of the mutant p53 [14,15]. Unlike other tumor suppressor proteins, mutant p53 may also undergo “gain of function” mutations, resulting in oncogenic transformation [16,17]. For example, it was shown that mice with mutant p53 developed more aggressive and metastatic tumors than mice with wild-type p53 [18,19]. Similarly, patients with mutant p53 developed more aggressive cancers than patients without the *TP53* mutation [20]. Mutations in the *TP53* gene and the dysfunction of a p53 protein lead to the genome instability, defects in DNA repair, the escape of cell cycle checkpoints, and uncontrollable division and disease progression. Since *TP53* is the most frequently mutated gene in cancer, targeting p53 mutations seems to be an attractive therapeutic strategy for the development of anticancer drugs. 

The P53 family also includes the p63 and p73 transcriptional factors. P53, p63 and p73 mainly consist of three domains, namely transactivation domain (TAD) on the N-terminal, the central DNA binding domain (DBD), and the C-terminal oligomerization domain (OD) [21,22,23]. TAD binds to either positive or negative transcriptional regulators, DBD binds to DNA, and OD is subjected to alternative splicing and post-translation modifications [21,22,23]. These domains are highly conserved between three different family members. The highest similarity between the group members was found in the DNA binding domain (~63%), while the transactivation and oligomerization domains demonstrated less similarity, with (~22%) and (~37%), respectively [24].

P73 plays a key role during the developmental stages, and is particularly important for neuronal differentiation [25]. It is believed that the role of p73 in cancer is very similar to p53, because this transcriptional factor was shown to activate the variety of common p53 targets [26]. Moreover, p73 acts as a tumor suppressor gene due to its role in the suppression of tumor-related processes, including cell cycle progression, genomic instability, the evasion of apoptosis and senescence, etc. [26] However, p73 can be also considered an oncogene due to its promotion of immune cell differentiation and stimulation of the immunosuppressive environment [27]. 

The P63 protein is highly expressed in progenitor layers of the epithelium and plays an essential role in the development of the limbs and epithelium [28,29]. For example, p63 knockout mice showed a lethal phenotype due to the absence of squamous epithelium and impaired limb formation [28,29]. Similarly, patients with p63 mutations are often presented with limb malformations and defects [30]. Depending on the isoform, p63 might play a different role in cancer development and progression, i.e., the p63 isoform with an N-terminal p53-homologous transactivation domain (TAp63) is believed to act as a tumor suppressor similar to p53 [31]. On the other hand, the p63 isoform, which lacks the TAp63 domain, may promote cancer development [31]. It was also shown that aggressive metastatic tumors were characterized by the loss of p63 expression, indicating its role in the suppression of cancer cell dissemination and metastasis [32]. Due to certain similarities in the function and structure of p53 family members, targeting p63 and p73 might be a promising therapeutic strategy for tumors with p53-altered or p53-deficient status [33]. 

The major therapeutic strategies to target p53 include the development of the drug candidates that are capable of restoring the function of wild-type p53 or those that are capable of suppressing or eradicating mutant p53. Although there are many small molecules that are currently being tested in clinical trials or at the advanced preclinical stage (e.g., mutant p53 activator COTI-2) [34,35], none of the molecules targeting p53 and other members of the p53 family has been clinically approved yet [36]. Since p53 is characterized by a large number of mutant alleles, targeting each allele with a different small molecule might not be efficient [37]. Moreover, since p53 plays an important role in normal cells, altering p53 function in normal tissues might result in serious side effects. For example, the MDM2 antagonist RG7112 demonstrated certain efficacy in patients with haematological malignancies and induced the transcriptional activation of p53-target genes in agreement with the proposed mechanism of action [38]. However, due to the role MDM2 plays in normal haematopoiesis, in addition to the suppression of leukemic cells, the treatment of RG7112 also caused the marrow suppression of normal progenitors, resulting in serious complications, such as sepsis, haemorrhage, and severe neutropenia [38]. While direct targeting of the p53 protein is hindered by its structural complexity, the indirect strategies based on acquired vulnerabilities or the oncogenic functions of p53 might be a more feasible therapeutic strategy.

## 2. Anticancer Metal Complexes Interfering with the p53 Protein

### 2.1. Zinc

#### 2.1.1. The Role of Zinc in the Structure and Function of p53

Several studies have revealed that the function of p53 depends on the presence of Zn ions. In a pioneering work, Cho et al. performed an X-ray diffraction analysis of a p53 protein (Figure 1) [39]. The DNA-binding domain of a p53 protein revealed two anti-parallel β-sheets with two motifs binding to the minor and major DNA grooves. Several loops at the DNA binding surface were shown to be stabilized by a tetrahedral coordination of a single Zn ion. This ion was coordinated by four amino acids, namely Cys-176, His-179, Cys-238 and Cys-242 [40,41,42]. Importantly, the binding to DNA could only occur in the presence of Zn ions [39], indicating the key role of Zn in the mechanism of action of a p53 protein [43]. In agreement, the treatment of the cells with Zn chelating agents resulted in the decrease of p53 activity [44]. 

The crystallographic analysis of mutated p53 analogues revealed that the majority of mutations were associated with the structural changes in the amino acid sequence (residues 102–296) responsible for p53-DNA interactions [39]. The mutations were classified based on their proximity to the main four amino acids that are coordinated with the Zn ion and by the impairing effect of Zn, which may lead to the protein misfolding [45]. The most common mutation in the Zn binding domain is R175H, and this mutation is considered to be the most frequent missense mutation in cancer. This mutation typically leads to impaired Zn binding and the misfolding of the p53 protein [41,46]. At physiological conditions, the wild-type p53 is Zn-bound (holo DNA binding domain), while the mutant P53 R175H is Zn-free (apo DNA binding domain), which explains the unfolding of protein in mutant p53 R175H form [47,48].

In the absence of Zn ions, the DNA binding region is considerably less stable (ΔG_1_ = 10 kcal mol^−1^ for the holo DNA-binding domain and 6 kcal mol^−1^ for the apo DNA-binding domain [41,49,50]. Moreover, the Zn-free apo DNA-binding domain demonstrated weak DNA binding and poor transcriptional activity. Importantly, Zn ions were shown to reactivate even inactive recombinant forms of p53. In order to investigate the effects of Zn ions on the mutated p53, Cho et al. used a recombinant p53 protein which was obtained from PA1620 and PAb240 antibodies [39]. Similar to the wild-type p53, the folding of mutated p53 in the absence of Zn ions resulted in the formation of protein structures with the impaired DNA binding function. However, the addition of Zn ions at ≈100–200 μM triggered the refolding of the protein and the full reactivation of the mutated p53 protein [51]. On the contrary, the addition of Zn-chelating molecules caused the unfolding and deactivation of p53.

Several studies investigated the effects of Zn overload on the function of the p53 protein. Since the p53 DNA-binding domain has different opportunistic Zn-binding sites, the excess of Zn ions resulted in the detrimental effects on its structure and function. For example, in the case of the native DNA-binding domain, the excess of Zn has led to its aggregation and precipitation [52]. Similarly, the addition of the excess of Zn to the unfolded DNA-binding domain resulted in the folding arrest and precipitation [52]. On the contrary, the addition of Zn to the DNA-binding domain at the equimolar ratio yielded a Zn complex with the p53 tetramer, characterized by the correct conformation and efficient DNA binding [52].

#### 2.1.2. The Effects of Zn Supplementation

Since the p53 protein was shown to be Zn-dependent and its mutations were associated with chemoresistance, several studies explored whether Zn supplementation might result in the reactivation of mutant p53 and the restoration of the chemosensitivity of cancer cells [53,54,55,56]. The addition of ZnCl_2_ to either cisplatin or Adriamycin was shown to enhance the apoptosis of SKBR3 breast cancer cells harboring the R175H mutation, and U373MG glioblastoma cells harboring the R273H mutation [54]. Based on the results of various assays, it was revealed that the improvement of drug response was related to the restoration of the proper conformation and activity of the p53 protein. It should be noted that Zn supplementation resulted in the dissociation of the p53/p73 complex and allowed for p73 to bind to DNA, thereby initiating cancer cell apoptosis. Importantly, Zn supplementation also enhanced the efficacy of the chemotherapeutic drugs in vivo in U373MG tumor xenografts. The analysis of harvested tumors from drug-treated groups revealed that Zn supplementation increased the PAb1620-reactive (folded) phenotype, indicating that the p53 protein transitioned into a functional conformation [54]. D’Orazi et al. also studied the effects of the Zn supplementation on the efficacy of low doses of Adriamycin in wild-type p53 colorectal HCT116 cancer cells, as well as in colon cancer mouse xenografts [57]. It was shown that Adriamycin did not induce tumor regression unless ZnCl_2_ was added, which suggests that ZnCl_2_ was the key determinant in the activation of wild-type p53 to bind to target DNA sequences. The combination treatment allowed for the reduction of the dose of Adriamycin leading to the reduction of potential toxic side effects [57]. Subsequently, the same authors questioned whether the supplementation of ZnCl_2_ to cisplatin or Adriamycin might affect the immunogenicity of dying cells [55]. It was shown that only a combinatorial treatment, but not chemotherapy alone or ZnCl_2_ alone, stimulated dying RKO cancer cells to activate the maturation of dendritic cells. Subsequently, the activated dendritic cells not only eradicated apoptotic cancer cells, but also improved the self-immunosuppression by presenting tumor antigens to cytotoxic T cells [55].

#### 2.1.3. Zn Complexes

Aiming to rescue the function of a mutant p53 protein, Carpizo et al. tested 48,129 compounds from the National Cancer Institute (NCI) database on 60 cancer cell lines with a different p53 status [58]. The screening revealed that three compounds from the class of thiosemicarbazones, namely NSC319726 (ZMC1), NSC319725 and NSC328784 (Figure 2), preferably inhibited mutant p53 cell lines, in particular cell lines with a 175-allele-specific p53 mutant. It was shown that NSC319726 was able to restore the DNA binding properties of the p53 protein with an R175 mutation, as well as transactivate the MDM2 promoter, leading to the restoration of the MDM2 negative feedback loop. The in vivo efficacy of NSC319726 was also allele-specific [58]. Notably, the in vitro activity of NSC319726 increased by several fold when it was administered together with Zn^II^ salt, leading to the formation of Zn-ZMC1 complex (ZN-1) (Figure 2) [58]. The crystal structure of ZN-1 [48] and Zn^2+^ titration of ZMC1 [59] revealed the binding stoichiometry of 2:1 (2 ZMC1 ligands and 1 Zn center, respectively). The increase of cytotoxicity in the presence of Zn^2+^ ions may indicate that NSC319726 functions as a Zn metallochaperone (ZMC), i.e., a Zn chelator that buffers the level of intracellular zinc, thereby restoring the conformation and function of the mutant p53 protein. Contrary to uncoordinated Zn^2+^ ions, which might cause the misfolding of p53 and other proteins [60,61], ZMC1 was shown to provide the selective source of Zn only for the native p53^R175H^ DNA-binding site [62]. The K_d_ value of ZMC1 (30 nM) is 15-times higher than the K_d_ value of a native p53 binding site, and 33-times lower than the K_d_ value of a non-native binding site; therefore, ZMC1 is believed to selectively re-metallate p53^R175^ DNA-binding without the misfolding of p53 and other proteins, such as albumin in serum [48,59]. Subsequently, it was demonstrated that nitrilotriacetic acid (NTA) chelated Zn^2+^ ions with a similar K_d_ value as ZMC1, and efficiently re-metallated the p53^R175H^ binding site, which is in agreement with the hypothesis [59]. However, the activity of NTA in cells was lower than the activity of ZMC1 [59]. 

Loh et al. investigated the mechanism of ZMC1- and NTA-mediated Zn^2+^ transport [48]. It was shown that HEK293 cells that were treated with ZMC1 (but not NTA) were characterized by the increased intracellular concentration of Zn^2+^ free ions. Additional experiments with a Zn-sensitive fluorescent probe revealed that even though both ZMC1 and NTA could bind to Zn^2+^ ions with a similar affinity, only the Zn complex of ZMC1 (ZN-1) could cross cellular membranes, possibly due to its neutral charge. It was suggested that inside the cells ZN-1 was protonated, resulting in the release of free Zn^2+^ ions and neutral ZMC1. To confirm that ZMC1 functions as a Zn ionophore, the authors investigated the metalation of mutant p53^R175H^ by ZMC1 in complete cell culture media with and without Zn chelators. It was shown that in the presence of Zn-chelating resin, the refolding of mutant p53 was abrogated, suggesting that extracellular Zn is important for the mechanism of action of ZMC1 [48]. 

Besides acting as a selective source of Zn, ZN-1demonstrated excessive ROS production via Fenton chemistry [59,63]. The activation of ROS resulted in a stress response, the transactivation of p53^R175H^ by phosphorylation, and apoptotic factor transcription [47,64]. Unfortunately, the re-metallation of mutant p53 can be considered a transient event, since after the increase of Zn concentration and the ‘switch on’ of mutant p53, the cellular homeostasis mechanisms were shown to lower the Zn level and ‘switch off’ p53 activation [47,62].

Despite the ability of ZnCl_2_ to restore a functional conformation of a p53 DNA-binding site [54,65,66,67], its effects were hindered by its poor intracellular accumulation [48]. Therefore, it was hypothesized that pre-formed Zn^II^ complexes with various lipophilic ligands might exhibit improved antitumor effects through a similar mechanism of action. For example, the pre-formed ZMC1-Zn complex, ZN-1 (Figure 2), was indeed more cytotoxic than ZMC1 against the epithelial ovarian TOV112D cancer cell line and more efficient in the reduction of pancreatic tumors in genetically engineered mice with a Zn-deficient allele (p53^R172H^), while having no effect on mice with non-Zn-deficient alleles (p53^R270H^) [68]. The reasons for the improved activity of ZN-1 in comparison with ZMC1 include the delivery of the optimal ratio of ZMC1 to Zn (2:1), as well as the decreased binding of the ZMC1 to other endogenous metals, such as Cu or Fe [68]. 

Another pre-formed heteroleptic Zn^II^ complex **1** with 4,4′-bis(hydroxymethyl)-2,2′-bipiridine (Figure 2), curcumin and chlorido ligands also exhibited cytotoxic effects in cancer cell lines with a mutant p53 status, namely SK-BR-3 breast cancer cells with an H175 mutation and U373 glioblastoma cells with an H273 mutation, but not in a cancer cell line with wild-type p53 [69]. It was shown that complex **1**, but not curcumin alone, could efficiently reactivate mutant p53 and enable p74 recruitment onto target promoters, resulting in the activation of apoptosis. As expected, complex **1** induced a conformational change in R175H and R273H mutant proteins and did not affect the native p53 conformation in cells with wild-type p53. Importantly, the complex **1**-mediated reactivation of p53 target genes was observed not only in vitro, but also in vivo in an orthotopic glioblastoma U373 model. The use of curcumin, which is characterized by intrinsic fluorescence, allowed for the visualization of the localization of **1** in tumor tissues [69]. 

Subsequently, the comparison of the p53 levels in SK-BR-3 cells (R175H mutation) and HCT116 (wild-type p53) treated with equimolar concentrations of complex **1** revealed the time-dependent decrease of p53 levels in SK-BR-3 cells and the time-dependent increase of p53 levels in HCT116 cells, without affecting p53 mRNA levels [70]. It was revealed that the reactivation of the mutant p53 protein by complex **1** triggered the process of autophagy, leading to the degradation of remaining mutant p53 proteins. In agreement, the inhibition of autophagy by specific autophagy inhibitors or the inhibition of p53 transactivation by pifithrin-α abrogated the degradation of mutant p53 proteins [70].The inhibition of ER stress impaired the autophagy induced by complex **1**, as well as the degradation of the mutant p53 protein [71]. Therefore, it was concluded that the prevalence of the wild type over the mutant p53 function was dependent on the ability of complex **1** to shift the balance between folded and misfolded p53 proteins and to trigger UPR activation [71].

Radulović et al. prepared a series of metal complexes with 2-formylpyridine selenosemicarbazone, including Zn complex **2** (Figure 2), and investigated their effects on p73 expression in a panel of cancer cell lines and vascular endothelial cells [72]. Complex **2**, as well as the uncoordinated ligand, upregulated p73 expression in HeLa, MDA-MB-361, EA.hy 926 and MS1 cells. Additionally, it was shown that complex **2** interfered with the cancer cell cycle and triggered cytochrome C release, leading to the induction of apoptosis [72]. 

Besides small Zn-containing molecules, several groups investigated the effects of Zn-containing nanoparticles. For example, photothermal Zn-doped Prussian blue nanoparticles were designed to release Zn^2+^ ions inside cancer cells, thereby triggering apoptosis and the autophagic degradation of a mutant p53 protein [73]. Similar effects were reported for Zn-Fe nanoparticles, which induced the degradation of p53 proteins with different mutations, but not the degradation of wild-type p53 [74]. 

### 2.2. Copper

#### 2.2.1. The Role of Copper in the Structure and Function of p53

Hainaut et al. investigated the effects of Cu binding on the conformation of a p53 protein and its DNA-binding activity [75,76]. It was shown that physiological concentrations of Cu^2+^ ions at 5–20 μM affected the conformation of a wild-type p53, thereby negatively affecting its DNA-binding [76]. The Cu^II^-mediated inhibition of p53 DNA-binding activity was prevented by the use of a Cu^I^-chelating agent, and was not affected by scavengers of ROS, suggesting that the DNA binding activity of a p53 protein might be dependent on Cu^II^/Cu^I^ redox transformations [76]. 

In a follow-up work, MCF7 breast cancer cells characterized by high levels of p53 protein and functioning p53 signaling were treated with a metal chelator, pyrrolidine dithiocarbamate (PDTC) [76]. The treatment did not cause cytotoxicity, but resulted in the down-regulation of p53 and a decrease in its DNA-binding activity. Additional mechanistic studies revealed that the down-regulation of p53 was linked to the significant increase of intracellular Cu levels, but not Zn or Fe levels [76]. It was shown that PDTC could selectively bind Cu^2+^ ions from cell culture media and transport them inside the cells, where they would undergo redox transformation to Cu^+^ ions and eventually bind to the p53 DNA-binding domain. The Cu-mediated down-regulation of the p53 protein prevented the p53 activation by DNA damage-dependent signals and delayed the cytotoxic effects caused by the oxidative stress inducers, such as H_2_O_2_ [76]. It should be noted that cellular Cu homeostasis is tightly regulated, and the imbalance of Cu levels was shown to cause the activation of cell death mechanisms in both cancer and healthy cells [77]. However, cancer patients are typically characterized by increased intracellular Cu levels [78], and it is tempting to speculate that the careful fine-tuning of the structure and physicochemical properties of Cu-based anticancer drug candidates might result in a certain degree of selectivity to cancer cells [77]. Given the role of Cu ions in the DNA-binding function of p53, several research groups sought to investigate whether Cu complexes can cause cancer cell death through the modulation of p53 expression. 

#### 2.2.2. Cu Complexes

Various structurally different Cu complexes were shown to be highly cytotoxic in different cancer cell lines [77,79]. Since intracellular Cu balance is tightly regulated, the fatal overload of bioavailable Cu leads to cancer cell death [77]. The Cu-induced cell death mechanism was recently termed cuproptosis [80]. It was suggested that cuproptosis might be regulated by p53 [81]. Besides cuproptosis, the mechanism of action of Cu complexes is typically linked to the induction of ROS based on the Fenton-like mechanisms [82]. There are several Cu complexes whose anticancer effects were investigated in the p53 context. The Schiff base Cu^II^ complex **3** (Figure 3) was shown to efficiently reduce the proliferation of MCF7 breast cancer cells in comparison with the uncoordinated ligand and CuCl_2_ [83]. Barlev et al. investigated whether the antiproliferative effects of **3** were related to the activation of p53 [83]. An immunoblot analysis illustrated the significant upregulation of p53 in **3**-treated cells, but not in ligand-treated cells. Besides increasing p53 protein levels, **3** also stimulated the increased expression of p53-related genes, such as MDM2, p21/CDKN1A, and PUMA. A comparison of the antiproliferative and apoptotic effects of **3** in both MCF7 p53wt and MCF7 p53^−/−^ tumor cells revealed that the antiproliferative activity of **3** was slightly higher in p53-positive MCF7 cells, while apoptosis was more pronounced in p53-negative cells. Therefore, even complex **3** was shown to affect the transcription and translation of p53, and some of its anticancer effects might be p53-independent [83].

A series of Cu^II^ complexes with a 2-pyridine-thiosemicarbazone ligand **4**–**8** (Figure 3) activated p53 expression, as reflected by western blotting and the induction of ROS production in the B-cell lymphoma family-2 [84]. Complex **7** demonstrated the highest cytotoxicity in a sub-micromolar concentration range and caused changes in mitochondrial membrane potential, leading to caspase-9/3 activation. Western blotting of MGC80-3 cells treated with complex **7** revealed that the protein levels of Apaf-1, Bad, Bax, and Cyt C significantly increased, while Bcl-2 levels decreased, indicating the induction of apoptosis. 

Harmse and de Koning et al. prepared a small library of Zn and Cu complexes with pyridine-substituted imidazo [1,2-*a*]pyridines (Figure 3) and investigated their activity against several breast and colon cancer cell lines in addition to leukemia cell lines [85,86]. While uncoordinated ligands and Zn complexes were characterized by poor anticancer activity, their Cu^II^ analogues were cytotoxic in a low micromolar and sub-micromolar concentration range. It was shown that the most cytotoxic Cu^II^ complexes **9**–**11** induced caspase-dependent apoptosis, followed by the induction of ROS and the disruption of mitochondrial membrane potential [86]. The expression of **11** apoptosis-related proteins was measured using a proteome apoptosis array in HT29 cells, since they are characterized by the high expression of p53, as well as its phosphorylated forms, due to the gain-of-function R273H mutation. It was shown that **9**–**11** significantly reduced the expression of phosphorylated forms of p53, phospho-p53 (Ser392), and phospho-p53 (Ser46), and the most considerable decrease of expression was observed for phospho-p53 (Ser15). These findings might imply that the investigated Cu^II^ complexes caused the loss of stability of the gain-of-function mutated p53 protein. The authors suggested that the inhibitory effects of **9**–**11** on p53 phosphorylation might be linked to the inhibition of ataxia-telangiectasia mutated (ATM) kinase [86], which is involved in the mediation of p53 phosphorylation [87].

Another Cu-based complex **12** with a mitochondria-targeting ttpy-TPP ligand (ttpy-TPP = 4′-p-tolyl-(2,2′:6′,2″-terpyridyl)triphenylphosphonium bromide) (Figure 3) was shown to cause the death of hepatocellular carcinoma, breast, lung and other types of cancer cells through the accumulation in mitochondria, leading to extensive mitochondrial damage [88,89]. Complex **12** promoted ROS induction, leading to the activation of p53 and its mitochondrial translocation, as reflected by the increased mitochondrial p53 levels. In agreement, the addition of an ROS-quenching reagent inhibited the increase of mitochondrial p53 levels while maintaining its cytoplasmic levels. The mitochondrial translocation of p53 was significantly reduced in Dynamin-related protein 1 (Drp1) knockdown cells, indicating the crucial role of Drp1 in p53-dependent mitochondrial damage. The flow cytometry experiments in wild-type p53, p53-null, and p53-mutant cells revealed that **12** induced higher apoptotic rates in wild-type p53 cells, suggesting that apoptotic cell death was p53-mediated [89].

Höti et al. proposed an interesting strategy based on the conjugation of Cu^II^- and Sn^IV^ fragments resulting in the formation of heterometallic complex **13** (Figure 3) [90]. According to the design, the Cu^II^ fragment was expected to specifically bind to N7-guanine DNA nucleobases [91], leading to strand breaks, while the Sn^IV^ fragment could potentially bind to the DNA phosphate backbone. Thus, two metallic fragments were expected to act in an additive manner. As a result, complex **13** caused serious DNA damage in HeLa cells, leading to a sharp increase in the levels of Bax and Bak and a decrease in the levels of the anti-apoptotic factor Survivin. HeLa cells can be considered as p53-defective, since p53 underwent continuous degradation by the human papilloma viral oncoprotein E6. However, the treatment of HeLa cells with complex **13** resulted in the significant increase of p53 protein expression together with the concurrent increase of p21. On the contrary, the treatment of the p53-deficient H1299 cell line (p53^−/−^) with **13** was not associated with any upregulation of p53 or p21, or changes in Survivin expression. Another experiment demonstrating the p53-dependent mechanism of action of complex **13** was conducted on the p53-deficient H1299 cell line (p53^−/−^) that was infected with either wild-type p53-expressing adenovirus or a mock GFP. It was shown that upon treatment with **13**, only H2199 cells that were infected with wild-type 53-expressing adenovirus were characterized by the release of cytochrome C and Smac/DIABLO from the mitochondria, while neither of these mitochondrial proteins was detected in the cytosolic fraction of p53-deficient cells with mock GFP. Importantly, **13** exhibited significant antiproliferative activity against tumor development in tumor-bearing rats, with minimal interference in their normal physiological function [90].

### 2.3. Iron

#### 2.3.1. The Role of Iron in the Structure and Function of p53

Fe cations were also shown to affect p53 expression, but indirectly. The statistical data obtained in the course of the treatment of various oncological malignancies demonstrated an obvious trend: an increased concentration of Fe in the cell increased the risk of cancer formation from 20 to 200 times [92]. Subsequently, a more detailed study on mice showed that the increase in the concentration of Fe in the blood correlated with the increase in the expression of human homeostatic iron regulator protein (Hfe), as well as the decrease of the p53 levels [93]. Hemin, which is a complex of the Heme protein with Fe, also suppressed the expression of p53. In contrast, various Fe chelators, as well as substances that suppress the expression of Heme, resulted in the upregulation of p53 expression. Thus, it was Hemin, and not Fe or Heme, that was shown to suppress p53. The proposed assumption was directly confirmed upon the isolation and subsequent mixing of pure p53 and Hemin, which resulted in the formation of the stable p53-Hemin complex.

However, some other studies reported contrasting observations with respect to Fe and p53 [94]. The authors described a number of experiments in murine and human cell lines using various Fe salts and a chelating agent: deferoxamine. The obtained results revealed a strong inverse relationship between Fe concentration and Mdm2. Thus, it was suggested that with an increase in the concentration of Fe, the concentration of free Mdm2 decreased, which, in turn, could lead to the destruction of the Mdm2-p53 complex and the activation of the target protein. Based on the described data, an increase in Fe concentration should lead to an increased expression of p53 [94]. Additionally, some studies reported that p53 affected the expression of hepcidin, thus being a regulator of Fe concentration itself [95]. Further studies showed that p53 also influenced other Fe regulators in the cell, such as ISCU and FDXR [96]. Hence, complex indirect interactions between Fe cations and p53 remain the subject of discussion and require additional research.

#### 2.3.2. Fe Complexes

The studies on Fe complexes inducing p53-mediated cell death are very scarce. For example, Chen et al. prepared Fe^II^ polypyridyl complexes **14**–**18** and investigated their effects in a panel of cancer cell lines (Figure 4) [97]. Complex **14** entered MCF7 breast cancer cells through transferrin receptor-mediated endocytosis and induced significant ROS production and DNA damage, leading to a cell cycle arrest in the S phase. Western blotting revealed that complex **14** caused an increase of p53 and p21 protein levels and a marked increase of phospho-p53(Ser15) levels. Similar to Cu complexes **9**–**11**, p53 phosphorylation induced by complex **14** was linked to the inhibition of ATM kinase. Additionally, the levels of phosphorylated *ATR* kinase were also increased, suggesting the involvement of a p53-dependent ATM/ATR signaling pathway.

To elucidate the relationship between the lipophilicity of the complexes, their cellular localization and molecular mechanisms of action, the pyridyl ligand in complex **14** was replaced by the more lipophilic phenanthroline and dipyridophenazine ligands [97]. Interestingly, while complex **14** accumulated in the nucleus, complex **15** was localized in the cell cytoplasm. Contrary to **14**, complex **15** and **16**–**18** did not cause significant DNA damage, but induced cell cycle arrest at the G_0_/G_1_ phase, which was in agreement with their differential intra-organelle accumulation. Complexes **16**–**18** caused the upregulation of p15, p18, p21, and p27 protein levels. These results might indicate that the mechanism of action of the more lipophilic complexes **15**–**18** might be both p53-dependent and p53-independent. The role of a metal center on p53 activation might be assessed by the comparison of tris(1,10-phenanthroline)Fe^III^ complex **15** and its structural La^III^ analogue (KP772) [98]. KP772 induced the expression of p53 and p21Waf1 in A549 cells with a wild-type p53 status. However, the subsequent comparison of the effects of KP772 in p53-null Hep3B-cells and its p53-transfected analogue (Hep3B/p53) revealed no significant differences in its activity, suggesting that, in contrast to Fe^III^ complex **15**, KP772 might act largely through p53-independent mechanisms [98].

### 2.4. Other Metals

#### 2.4.1. The Role of Other Metals in the Structure and Function of p53

Unfortunately, the effect of other metals on p53 was described extremely poorly. There is some data on protein binding by Co cations, which, like Zn, caused the partial renaturation of the p53 protein, but at much higher concentrations [48]. It was shown that the effect of 125 micromoles of Co cations was similar to the effect of 5 micromoles of the Zn cations. Additionally, the p53 protein could bind to Cd, resulting in the inhibition of its activity [99]. However, Cd is a highly toxic metal with a limited medicinal use; therefore, we will not cover this metal in detail here.

#### 2.4.2. Ru Complexes

Ru anticancer complexes have attracted a lot of interest in the cancer research field [100]. Initially, the idea to use Ru to treat cancer was based on some degree of similarity between Ru and Fe (Group 8 in Periodic Table), enabling the use of Fe transport proteins, such as transferrin, for the intracellular delivery of Ru complexes [101,102]. However, it was later revealed that the transport of Ru complexes through cellular membranes most likely occurred through interactions with albumin, rather than transferrin and other Fe proteins [103,104]. Nevertheless, Ru complexes demonstrated various advantages over classically used Pt-based chemotherapeutic agents, including reduced toxicity and resistance. Moreover, the structure of Ru complexes can be easily modified via straightforward chemical methods, which is in agreement with the required drug design. Many studies investigated the role of various structurally different Ru in the activation of p53 pathway.

Ru^II^ complexes **19** and **20** containing bis-benzimidazole derivatives (Figure 5) were designed as radiosensitizers for use in cancer treatment [105], since nitrobenzimidazoles were shown to sensitize hypoxic cells to radiation [106]. As expected, the cytotoxicity of both complexes against human melanoma A375 cells was 1.6–2.4 higher when it was combined with X-ray radiation. The mechanism of action of compound **20** was linked to ROS production due to its interaction with cellular glutathione, leading to the formation of DNA breaks, p53 activation, and cell cycle arrest at the G_2_/M phase. Another structurally similar Ru complex **21** (Figure 5) was shown to induce p53-mediated DNA damage and mitochondrial dysfunction [107]. Importantly, the drug-induced sensitization of A375 cells to radiation also occurred in a p53-dependent manner [105]. The co-treatment of cancer cells with complex **19** and X-ray markedly enhanced p53 expression, p53 phosphorylation at Ser15, and decreased the expression of MDM2 in comparison with only complex **19**. Moreover, p53 activation was associated with the activation of the extrinsic death-related apoptosis pathway, since the overexpression of p21 coincided with the decrease of protein levels of death receptor 5 (DR5) [105].

The effects of Ru-chrysin complex **22** (Figure 5) were investigated not only in vitro, but also in vivo in female Sprague–Dawley rats with 7,12-dimethylbenz(α)anthracene-induced mammary cancer [108]. When MCF7 breast cancer cells were treated with **22** for 24 h, the notable upregulation of p53 expression was observed, which coincided with the downregulation of mTOR and VEGF expression. Importantly, the p53 expression was quantified in the mammary cancer tissues of control animals, as well as animals treated with complex **22**, RuCl_3_, or chrysin alone. The p53 expression in tissues from the RuCl_3_ and chrysin groups was similar to the p53 expression levels in the untreated group. However, the analysis of tumors from the **22**-treated group revealed a two- to three-fold higher p53 expression than in the untreated group. Complex **22** was also shown to induce DNA fragmentation, apoptosis, and the arrest of cellular proliferation. It should be noted that complex **22** is a rare example of a Ru complex whose effects on p53 were verified and quantified in vivo [108]. Subsequently, the same authors prepared a structurally similar analogue of complex **22**, where Ru^II^ was replaced with an oxo-V^IV^ fragment. The structurally similar V complex did not show any differences with complex **22**, indicating that while the presence of the metal is important, it might only play a structural role [109].

Gaiddon et al. prepared a small series of structurally different Ru^II^ complexes and evaluated their cytotoxicity in an A172 glioblastoma cell line (Figure 5) [110]. Complexes **23** and **24** revealed the highest anticancer activity in a low micromolar range among the tested complexes, in addition to the ability to induce apoptosis. They demonstrated the rapid upregulation of p53 protein levels, followed by the activation of p21 and Bax. To investigate whether the pro-apoptotic effects of **23** were p53-mediated, the authors tested their effects in ΔNp73-A172 cells (dominant negative p73 isoform inhibiting p53 and p73 activity) and in p53DD-A172 cells (dominant negative p53 isoform, containing the truncated form of p53) [110]. It was shown that the extent of apoptosis induced by **23** in modified cell lines was only slightly reduced in comparison with wild-type A172 cells, indicating that these complexes might induce both p53-dependent and p53-independent mechanisms. On the contrary, the pro-apoptotic effects of cisplatin were completely abolished in the modified cell lines. Additionally, the cell cycle inhibitory effects of **23** and cisplatin were compared in TK6 cells (p53^+/+^) and NH32 cells (p53^−/−^) [110]. Whereas the activity of cisplatin was abrogated in NH32 cells, the activity of **23** was comparable in both cell lines, which is in agreement with the previous findings. Similar observations were reported for the series of Ru^II^-Arene Schiff-base (RAS) complexes, including complex **25**, whose cytotoxicity in HCT116 cells was not changed upon the addition of the p53 inhibitor, pifithrin-α, while the cytotoxicity of a p53-dependent chemotherapeutic drug oxaliplatin was reduced by at least three-fold [111]. The ability of Ru complexes to induce apoptosis in a p53-independent manner might be useful in overcoming the resistance mechanisms to classically used chemotherapeutic agents, including cisplatin, oxaliplatin, etc.

It is known that the conjugation of several structurally similar or different metal fragments might lead to the polynuclear scaffolds with improved anticancer properties [112]. The improvement of the biological effects might be related to the synergistic mechanism of action of metal fragments or the enabling of long-range and more efficient interactions with the intended biomolecular target [112]. There are several reports on the investigation of polynuclear Ru complexes in the context of p53 activation. Bimetallic Ru^II^ complexes **26**, **27** and **28** comprised of two half-sandwich Ru^II^ fragments that coordinated to either a bis-salicylaldimine ligand [113] or a bis-pyrimidine-derived ligand [114] (Figure 5) demonstrated p53-dependent anticancer effects. A western blot analysis revealed the significant increase of p53 protein levels in drug-treated cancer cells, as well as the upregulation of the protein expression of p63 and p73 (measured for complexes **27** and **28**) or p21 and p15 (measured for complex **26**). Additionally, the increase of mRNA levels of p53-relevant target genes, such as Bax, PUMA, and NoxA, was observed [113,114]. It should be noted that complex **28** also induced cell cycle arrest at the S and G2/M phases, and caused a considerable decrease of the protein levels of *c*-MYC, which is one the key oncogenes that is responsible for cancer development and progression [115]. All studied bimetallic Ru complexes induced apoptosis and inhibited cancer cell migration and invasion.

Bezerra et al. prepared a heterobimetallic Ru^II^-Fe^II^ complex **29** based on the Ru-piplartine fragment and bis(diphenylphosphino)ferrocene and its monometallic Ru^II^ analogue **30** with 1,4-bis(diphenylphosphino)butane (Figure 5) [116,117]. It should be noted that piplartine, which is a naturally occurring compound found in the fruit of long pepper, was previously shown to induce the activation of p53 signaling [118]. As expected, Ru-piplartine complexes **29** and **30** induced p53-mediated apoptosis, which was considerably reduced in the presence of the p53 inhibitor pifithrin-α. Due to the incorporation of two metal centers into the structure of the metal complex, the cytotoxicity of **29** and **30** was higher in comparison with an uncoordinated piplartine ligand. Moreover, both **29** and **30** were effective in reducing the tumor burden in an HCT116 mouse xenograft model [116].

Se-containing Ru^II^ complex **31** (Figure 5) was designed to potentiate the cytotoxic effects of natural killer (NK) cells in prostate cancer [119]. The design was based on the observation that the conjugation of Ru and Se fragments resulted in the enhancement of cytotoxicity due to the stimulation of ROS-mediated ER stress [120], which is important in the immune modulation. In agreement with the design, the treatment of PC3 and LNCAP cancer cells with subtoxic doses of **31** effectively increased the lytic capacity of NK cells up to 2.5-fold compared to NK cells alone. More importantly, low concentrations of **31** enhanced the tumor-killing potential of NK cells. Mechanistic studies revealed that the sensitization effect of **31** was mainly dependent on TRAIL/TRAIL-R and Fas/FasL signaling. In addition, **31** showed the potential to activate and cooperate with NK cells in vivo without inducing toxicity to major organs. The investigation of the potential mechanism of complex **31**-mediated NK cell activation revealed that **31** induced DNA damage in cancer cells by engaging ATM and ATR key transducers, followed by the activation of Chk1 and Chk2 kinase checkpoints and p53 [119]. In turn, p53 induced the expression of various ligands for NK cell-activating receptors [121].

#### 2.4.3. Au, Ag and Pd Complexes

Au complexes containing N-heterocyclic carbenes have emerged as an alternative to classically used chemotherapeutic agents due to their excellent anticancer activity in a broad panel of cancer types, as well as their high stability in biological conditions and efficient intracellular accumulation [122,123,124,125]. The mechanism of action of Au complexes has been extensively investigated, and is believed to involve the inhibition of thioredoxin reductase, ROS production, and the induction of mitochondrial dysfunction and ER stress [123]. Cheng et al. performed a detailed investigation on the relationship between the cytotoxic effects of an Au^I^-NHC complex **32** (Figure 6) and the p53 status of different cancer cell lines [126]. The anticancer effects of **32** were investigated in human colorectal cancer cell lines with three different p53 variants, namely wild-type HCT116, HCT116 p53^−/−^, and HT29 (mutant R273H). The complex demonstrated the highest cytotoxicity in the cell line with unaltered p53 status and the lowest activity in HT29 cell lines with the R273 mutation. This observation indicates the involvement of the p53 protein in the mode of action of **32**. In agreement with this, when HCT116 cells were transfected with plasmids carrying different p53 mutations, the cytotoxic effects of **32** were abrogated. As expected, the mechanism of action of **32** was linked to the induction of ROS, interfering with the cell cycle, and leading to apoptosis. These effects were the most pronounced in a cell line with a wild-type p53, followed by the cell line with null p53, and almost no effects were observed in the HT29 cell line with the mutant p53. The activation of the p21 cell cycle checkpoint, as well as PARP cleavage, was detected in all cell lines, independent of p53 status. On the contrary, the treatment of both HCT116 and HT29 cell lines with complex **32** resulted in the decrease of p73 expression, including the expression of the Tap73 and ΔNp73 isoforms. In agreement with this, the cytotoxicity of complex **32** was not attenuated in cells with the transient knock down of p73, indicating the p73-independent mode of action. The knockdown of p21 in HCT116 wild-type and HCT116 p53^−/−^ cells resulted in the loss of cytotoxicity of **32** in wild-type cells and the stimulation of cell viability in null p53 cells. To conclude, the mechanism of action of Au^I^-NHC complex **32** was indeed dependent on the p53 status: while complex **32** induced apoptosis in cell lines with a wild-type p53 status, the mechanism of action in cell lines with an altered p53 status was related to the cell cycle interference [126].

Similar findings were reported for another Au^I^-NHC complex, **33** (Figure 6), which was cytotoxic in a low micromolar concentration range in the HCT116, HepG2, A549, and B16F10 cell lines [127]. Treatment of B16F10 melanoma cells with **33** stimulated the activation of p53-dependent apoptosis, characterized by the upregulation of p53 and p21 expression, as well as p53 phosphorylation. The co-incubation of the cells with pifithrin-α reduced the expression levels of p53 and completely inhibited the expression of p21, but did not affect the ROS production. However, the cytotoxicity of **33** in the presence of pifithrin-α was reduced but not completely abrogated, again indicating that this type of complex might act in both a p53-dependent and a p53-independent manner. The p53-dependent effects of **33** were also confirmed in vivo [127]. The histopathological analysis of tumor tissues isolated from drug-treated Balb/C mice bearing B16F10 tumors revealed the translocation of p53 to the nucleus and the significant upregulation of p21 levels, as well as the inhibition of anti-apoptotic NF-κB, VEGF, and MMP-9 proteins.

Ong and Li et al. studied the effect of the metal center on the cytotoxicity of complexes **34**–**36** with an amino-linked heterocyclic carbene [128] (Figure 6). The cytotoxicity of Pd complex **36** in MCF7 and MDA-MB-231 breast cancer cell lines was at least 2.5-times higher than the activity of Au complex **35**, and at least 4-times higher than the activity of Ag complex **34**. In comparison, the cytotoxicity of Au complex **35** in the glioblastoma U-87 MG cell line was at least 8- and 20-times higher than the cytotoxicity of **36** and **34**, respectively. It should be noted that it was not possible to deduce structure-activity relationships, since three complexes were structurally different and varied by metal fragments and the ligand. Complex **34** induced the dose- and time-dependent activation of p53 and p-p53(Ser15) in U-87 MG cells, in addition to stimulating PARP cleavage. In contrast to previously described Au-NHC complexes, treatment of cells with **34** resulted in the dose-dependent decrease of p21 expression. It is known that brain tumors are often characterized by the high levels of p21, which makes them more resistant to p53-dependent apoptosis [129]. Therefore, the inhibition of p21 expression levels might be beneficial for the treatment of U-87 MG glioblastoma.

#### 2.4.4. Pt and Pd Complexes

Pt compounds, such as cisplatin, oxaliplatin and carboplatin, represent an important class of anticancer agents due to their extensive use in chemotherapy [130,131]. However, their long-term treatment is hindered by the acquired resistance; moreover, many types of tumors are intrinsically resistant to Pt-based chemotherapy [132]. The causes of Pt resistance include reduced drug accumulation due to the altered drug transport mechanisms, increased drug detoxification, and alterations in the DNA repair and apoptotic cell death pathways [133,134]. There is excessive evidence that resistance to cisplatin could be linked to p53 mutations and the loss of wild-type p53 function [135,136]. However, there is no clear-cut relationship between the sensitivity of cancer cells to Pt treatment and their p53 status. For example, cisplatin-resistant ovarian 2780CP cells with a p53 mutation at codon 172 were characterized by the increased DNA damage tolerance in comparison with the cisplatin-sensitive ovarian A2780 cell line with a wild-type p53 gene [137]. As expected, cisplatin-treated A2780 cells demonstrated a dose-dependent increase of p53 expression, while cisplatin treatment in 2780CP cells did not cause the activation of p53. However, when 2780CP cells were irradiated by X-ray irradiation, the increase of p53 expression was observed, suggesting that p53 activity was at least partially retained [137]. These results indicated that the sensitivity of cancer cells to cisplatin was not only dictated by p53 status/function, but also other factors. In contrast to A2780 and 2780CP ovarian cancer cells, the effects of cisplatin in cisplatin-sensitive and cisplatin-resistant testicular cancer cell lines (Tera and Tera-CP, respectively) were associated with the activation of p53 [138]. On the other hand, when testicular cells with intrinsic cisplatin resistance (2012EP and Scha) were treated with cisplatin, the suppression of p53 and the activation of apoptosis were observed [138]. In human lymphoma cells, the mutations of the p53 gene were related to the decreased sensitivity of cancer cells to cisplatin [139]. However, the inactivation of wild-type p53 in MCF7 breast cancer cells led to the increase of their sensitivity to cisplatin treatment [140]. These contradictory observations indicate that the role of p53 in the cisplatin-mediated apoptosis is strongly dependent on the cell type [138].

Unlike cisplatin, oxaliplatin is clinically effective for the treatment of colorectal cancers; therefore, the activity of oxaliplatin was tested in 10 colorectal cancer cell lines and linked to their p53 status [141]. In general, oxaliplatin was significantly more active in cell lines with a wild-type p53 status (e.g., HCT116 or LoVo) than in cell lines with a mutated p53 status (e.g., HT29 or SW480). However, the V9P cell line with a mutated p53 status was at least 10-times more sensitive than HT29, SW480 or Isreco1 cells, indicating the involvement of p53-independent mechanisms [141]. Similar observations were reported by Perego et al., who analyzed the relationship between the cytotoxicity of oxaliplatin and the p53 status of several cisplatin-sensitive and cisplatin-resistant cancer cell lines [133]. Oxaliplatin demonstrated lower activity in cell lines with a p53 mutation and higher activity in cells with a wild-type p53. However, the activity of oxaliplatin in cisplatin-sensitive IGROV-1 cells and U2-OS cells differed by at least 10-fold, even though both cell lines were characterized by the presence of a wild-type p53 [133]. In general, the mechanisms of chemosensitivity and chemoresistance are multifactorial, in particular in cell lines with mutated p53, and therefore are exceptionally complex.

It is well-known that Pt-based chemotherapeutic agents are associated with severe toxicity, including nephrotoxicity, ototoxicity, neurotoxicity, etc. Since Pt drugs were shown to activate p53, leading to cancer cell apoptosis, it was investigated whether p53 might be also involved in the apoptosis of normal cells, such as renal tubular cells [142]. Dong et al. demonstrated that the activation of the ATM-Chk2-p53 pathway in cultured rat kidney proximal tubular cells was an early signal for tubular cell injury following cisplatin exposure [142,143]. In agreement with this, p53-deficient mice were at least partially protected from cisplatin-induced damage, as reflected by the improved renal function, and reduced apoptosis and tissue damage [144]. Similarly, the activation of the ATM-Chk2-p53 pathway was a major determinant for the onset of ototoxicity and hearing loss, as well as hair loss [145]. It was shown that the administration of cisplatin together with pifithrin-α to mice bearing triple-negative breast tumors with a wild-type or mutant p53 prevented the death of cochlear and hair cells. Importantly, pifithrin-α did not attenuate the efficacy of cisplatin treatment in wild-type p53 tumors, and improved the effects of cisplatin in mutant p53 tumors. In another work, Pt-induced peripheral neuropathy was linked to the mitochondrial accumulation of p53, resulting in mitochondrial damage [146]. The co-administration of pifithrin-μ to female mice prevented the formation of cisplatin-induced abnormalities in the mitochondria of dorsal root ganglia and the sciatic nerve [146]. These findings are important, since hearing loss, hair loss and peripheral neuropathy in patients can be potentially reversed or prevented by the use of p53 inhibitors during cisplatin treatment [145].

Zhu et al. proposed an elegant approach based on the chemical conjugation of cisplatin and chalcone, which was reported to act as an inhibitor of p53-MDM2 interactions [147]. The resulting Pt(IV) prodrug **37** was shown to release both components upon reduction in the cancer cell milieu. As expected, in cancer cells with a functional p53 complex, **37** was more effective than cisplatin. In contrast, the cytotoxicity of **37** was comparable to cisplatin in 53-null cancer cells. In agreement with the design, complex **37** demonstrated a dramatic increase of p53 expression in comparison with cisplatin [147].

Several stable Pt(II) complexes with chelating ligands were shown to activate the p53 signaling pathway, including a series of Pt(II)-tacrine complexes designed by Liang et al. [148]. The most cytotoxic complex, **38**, displayed excellent cytotoxicity in a low micromolar and submicromolar concentration range against a small panel of cancer cell lines, as well as marked in vivo efficacy comparable to cisplatin in a Hep-G2 xenograft mice model [148]. The anticancer activity of **38** was related to the activation of p53, p21 and p27 expression, leading to apoptosis. The percentage of apoptotic cells decreased at least 2-fold when p53-depleted cells were treated with **38**, indicating that apoptosis was at least partially p53-dependent. However, the activity of **38** in p53-depleted cells was still considerably high, suggesting that the mechanism of action of this complex is also based on the p53-independent mechanisms. The cytotoxicity of Pt(II) complex **39** based on the derivative of the cell penetrating peptide D-maurocalcine (D-MCa) was also only partially dependent on p53 [149]. Complex **39** caused dose-dependent p53 phosphorylation in human glioblastoma U87 cells, leading to apoptosis [149]. This complex was also shown to activate the intrinsic mitochondrial pathway, as well as the extrinsic TNF/TRAIL pathway.

Several Pt and Pd complexes, namely Pt(II) complex **40** [150] with a thiocarbohydrazone ligand and Pd(II) complex **41** [151] with a bis-imidazolium-based *N*,*N*′-bis-(salicylidene)-*R*,*R*-1,2-diaminocyclohexane ligand, were reported to significantly upregulate the p53 expression in various cancer cell lines; however, the dependence of the mechanism of action on the p53 has not been investigated. Another Pd(II) complex **42** with a terpyridine ligand and saccharinate counterion was investigated in vivo in an Ehrlich Ascites Carcinoma solid tumor model in comparison with cisplatin and paclitaxel, and their effects on p53 and other markers were assessed in tumor tissues using immunohistochemistry [152]. In this model all tested compounds caused the decrease of p53 and Bcl-2 expression, and the increase of Bax and Caspase-3 expression.

#### 2.4.5. Other Transition Metal Complexes

According to the analysis of the COMPARE algorithm from the National Cancer Institute, the mechanism of action of anticancer Ir^III^ complexes was based on the DNA interactions and the inhibition of protein synthesis [153]. Since the regulation of DNA function is largely exerted by p53, it might be interesting to study whether the effects of Ir^III^ complexes are p53-dependent. The organometallic Ir^III^ complex **43** (Figure 7) demonstrated excellent cytotoxicity in a sub-micromolar concentration range in a panel of cancer cell lines with different p53 status [154]. The quantification of Ir in isolated nuclear DNA fractions revealed that that the Ir-bound content in A2780 ovarian cancer cells (wild-type p53) treated with complex **43** was at least 13-times higher than in cells treated with cisplatin. In agreement with this, **43** caused higher levels of apoptosis and DNA fragmentation than cisplatin. The comparison of cytotoxicity in A2780 cells (wild-type p53) and HL-60 cells (mutant p53) revealed that cisplatin was at least two-times less active in cells with mutant p53, while **43** was equally cytotoxic. Similarly, complex **43** strongly blocked the G0/G1 phase in both cell lines, while cisplatin mostly blocked the S phase, and the effects were more pronounced in the wild-type p53 cell line [153].

Leung, Wang and Ma et al. developed a series of luminescent cyclometalated Ir^III^ complexes for the potential applications in the treatment of malignant melanoma [155]. It was hypothesized that Ir complexes might be able to interact and simultaneously monitor the S100B protein, which is expressed in the majority of melanoma tumors [156]. The expression of the S100B protein typically correlates with the grade of the melanoma [156]. It is known that S100B protein inhibits p53 pathway via interaction with the C-terminus of p53 protein, as well as via binding to the tetramerization domain of p53 protein [157]. Complex **44** (Figure 7) exhibited the inhibition of S100B/p53 at ca. 96%, and was identified as the lead compound within the series of 13 structurally similar cyclometalated Ir^III^ complexes. The substitution of Ir with Rh resulted in a marked drop of inhibitory activity and an alteration of the redox properties, suggesting the importance of the Ir metal center in the structure of complex **44**. In agreement with the excellent inhibition of S100B/p53, complex **44** demonstrated excellent cytotoxicity in A375 melanoma cells in a nanomolar concentration range (26.4 nM), as well as marked efficacy in B16F10 and A375 melanoma in vivo models. The authors engaged a cellular thermal shift assay (CETSA) to quantify the extent of binding of **44** to S100B. It was shown that **44** significantly stabilized S100B protein, but not the related S100A4 and hDM2 oncoproteins. Further experiments revealed that **44** preferentially bound to the C-terminus of p53 (97%) rather than its tetramerization domain (20%). Complex **44** inhibited S100B/p53 interactions and restored p53 function, as confirmed by the co-immunoprecipitation and the luciferase reporter assays, respectively. Following the inhibition of S100B/p53 interactions, cancer cells underwent p53-dependent apoptosis, associated with the upregulation of p21 and Bax expression both in vitro and in vivo [155].

V complexes have been widely investigated as potential therapeutics for the treatment of Type 2 diabetes and obesity [158,159]. However, they also found a potential application as anticancer agents due to their ability to interact with DNA, cause oxidative stress, and arrest the cell cycle [160]. For example, the oxovanadium complex **45** with a Shiff-base ligand (Figure 7) was investigated in MKN45 gastric cancer cells [161]. The analysis of the gene expression in drug-treated cells by the real-time PCR analysis revealed the upregulation of p53, GADD45, and CDC25 expression. These results were in agreement with the ability of complex **45** to induce cell cycle arrest at the G_2_/M phase. The accumulation of cells in the G2 phase prevented their entry into the division phase and repressed their unregulated proliferation. Similar observations were reported for another oxo-V^IV^ complex, **46**, with the 2-methylnitrilotriacetate ligand (Figure 7), which caused the cell cycle arrest at the G_2_/M phase, as well as the marked upregulation of p53 and p21 expression [162]. Vinklárek et al. tested two structurally unrelated V and Mo complexes, **47** and **48** (Figure 7), against a panel of 14 cell lines, including those with the wild-type p53 status (e.g., MOLT-4 lymphoblastoid leukemic cells) and p53-deficient cells (e.g., Jurkat lymphoblast T-cell leukemic cells) [163]. Both complexes induced the rapid and dose-dependent increase of p53 expression in MOLT-4 cells, as well as the phosphorylation of p53 at the Ser15 site. However, both complexes were similarly cytotoxic in cells with a different p53 status, suggesting that their anticancer mechanism at least partly does not depend on p53. Several other V complexes demonstrated similar effects on the cell cycle and p53 expression in different cancer cells [160]; however, these studies lack in-depth mechanistic investigation and will not be discussed here in detail.

#### 2.4.6. Complexes with p-Block Elements

The biologically active complexes with p-block elements, such as Ge, Sn and Sb, are less investigated than transition metal complexes. In contrast, Bi and its complexes found their application in the treatment of gastrointestinal disorders due to their excellent gastroprotective effects [164,165,166]. The potential use of Bi compounds as anticancer agents and radionuclides has also been documented [165,167]. Yang and Sun et al. prepared four binuclear Bi^III^ complexes **49**–**52** with modified 2-Acetyl-3-ethylpyrazine thiosemicarbazides (Figure 8) [168]. While the uncoordinated ligands were not cytotoxic, **49**–**52** demonstrated activity against various cell lines in a micromolar concentration range. The activity in all cell lines followed the same trend: **49** < **52** < **51** < **50** in correlation with their intracellular accumulation. The lead complex **50** arrested the cell cycle in the S-phase in human bladder T24 cancer cells by the regulation of cyclins and cyclin-dependent kinases. Western blotting revealed that T24 cells treated with **50** were characterized by the downregulation of Cdk2 and Cdc25A expression and the upregulation of p53 and p21 expression. Additionally, complex **50** induced ER stress and oxidative stress, in addition to stimulating the autophagy and death receptor pathways [168]. Sn-based complex **53** derived from propyl gallate and phenanthroline (Figure 8) effectively activated DNA strand breaks in MCF7 breast cancer cells in a dose-dependent manner [169]. As a consequence, the expression of a DNA damage marker, phosphorylated histone H2A.X (Ser139), was upregulated in drug-treated cells. Western blotting and fluorescent microscopy results revealed the dose-dependent phosphorylation of p53, suggesting that it might potentially mediate drug-induced DNA damage; however, further detailed mechanistic investigations have not been performed.

Interestingly, arsenic trioxide (As_2_O_3_), which is used for the treatment of acute promyelocytic leukemia (APL), demonstrated very pronounced effects on the stability of mutant p53 proteins [170]. The treatment of cancer cells with wild-type p53, such as MCF7, RKO, HCT116 and MEF, with arsenic trioxide resulted in the dose-dependent increase of p53 expression. On the contrary, HaCaT, SW480 and MIA PaCa-2, were characterized by the time- and dose-dependent reduction of p53 expression. This drug induced the proteosomal degradation of mutant p53 proteins with various mutations, including p53^R175H^, p53^H179Y/R282W^, p53^R248W^ and p53^R273H^. Additionally, As_2_O_3_ decreased the stability of mutant p53 proteins by blocking its shuttling between the nucleus and cytoplasm [170].

## 3. Conclusions

The P53 protein remains one of the most actively investigated cancer targets. While several organic small molecules demonstrated promising effects in preclinical studies and clinical trials, the direct targeting of the p53 protein is hindered by its incredible structural complexity and a large number of mutations. In this review, we put an emphasis on the dependence of p53 proteins on certain endogenous metals, in particular on Zn, which might be its unique vulnerability and can be exploited as a therapeutic strategy. Since the wild-type p53 protein is known to contain Zn in its active site, and its mutant forms are Zn-deficient, the Zn supplementation as a Zn salt or pre-formed Zn complex resulted in the conformational changes in the p53 protein and the restoration of its function. Therefore, the development of anticancer Zn complexes might be valuable for p53 targeting. However, we should keep in mind that the re-metallation of mutant p53 can be considered a transient event, and its function could be switched off by the cellular homeostasis mechanisms even after the restoration of its function. In comparison with Zn complexes, various structurally different complexes with non-endogenous metals, such as Ru, Ir, Au and others, induced anticancer effects through p53-dependent and p53-independent mechanisms. Hence, the development of the complexes based on the non-endogenous metals might be advantageous for battling the acquired p53 resistance.

What do we really know about p53-targeting metal complexes? The analysis of the literature revealed that this field remains largely unexplored. With the exception of Zn complexes, the majority of scientific studies on p53-dependent anticancer metal complexes did not include the information on their interactions with a p53 protein, and only demonstrated the effects of metal complexes on the p53 expression of cancer cells (Table A1). One should keep in mind that due to the critical role of p53 in the function of normal cells, the effects of p53-targeting metal complexes might be potentially toxic. For example, the toxicity induced by the clinically used drug cisplatin was linked to its activation of p53, but it could be prevented or reversed by the co-administration of p53 inhibitors. It would be interesting to study whether the p53-targeting metal complexes can induce p53 activation in normal tissues, and whether it can be reversed by the addition of p53 inhibitors, such as pifithrin-α. We believe that more in-depth mechanistic studies on metal complexes in the p53 context, e.g., the effects on the protein conformation and degradation, would be very desirable and could stimulate the preclinical development of metal-based drug candidates.

## Figures and Tables

**Figure 1 cancers-15-02834-f001:**
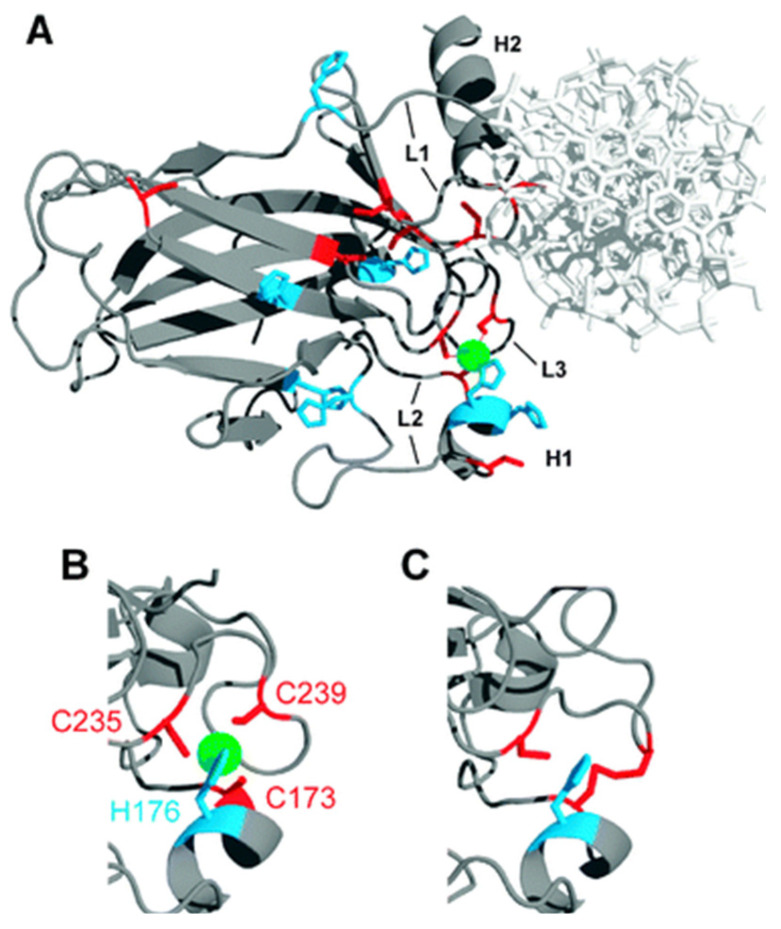
(**A**) The X-ray structure of the human p53 DNA binding domain. The green atom indicates a Zn ion; (**B**) The excerpt of the mouse p53 DNA binding domain. The green atom indicates a Zn ion; (**C**) The excerpt of the mouse p53 zinc-free DNA binding domain (apoDBD). The figure was reproduced from reference [43] with the permission of the publisher.

**Figure 2 cancers-15-02834-f002:**
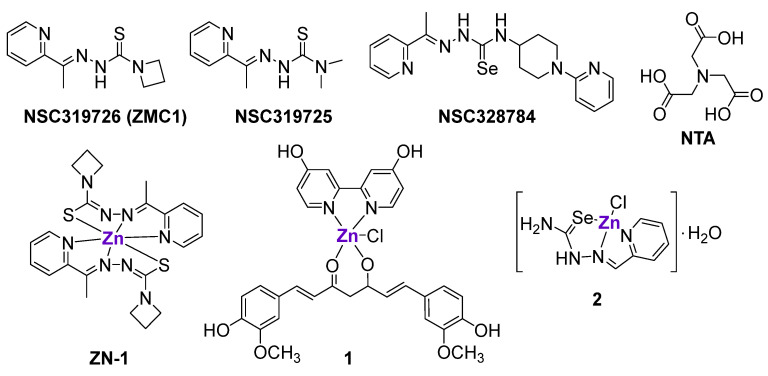
Zn-chelating ligands and Zn^II^ complexes interfering with the mutated p53 DNA binding site or inducing the upregulation of p53 family members.

**Figure 3 cancers-15-02834-f003:**
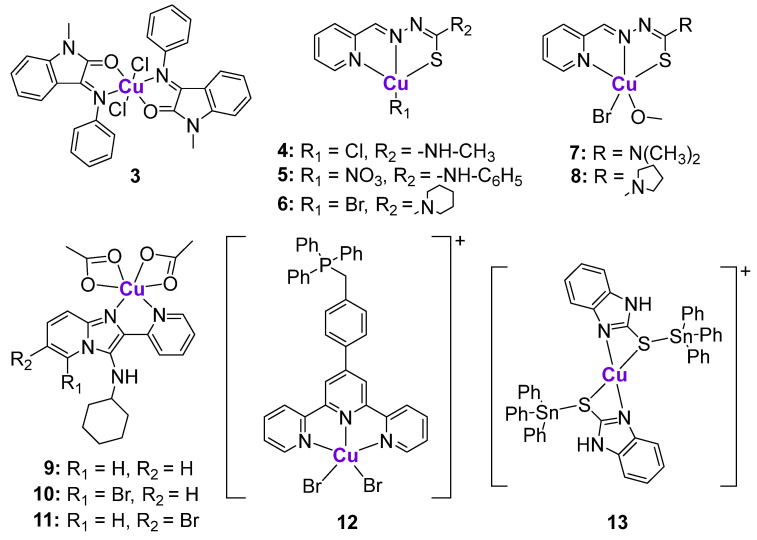
Cu complexes affecting p53 expression.

**Figure 4 cancers-15-02834-f004:**
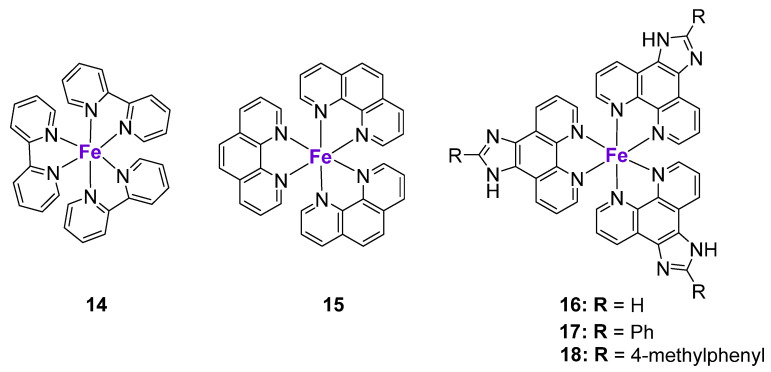
Fe complexes affecting p53 expression.

**Figure 5 cancers-15-02834-f005:**
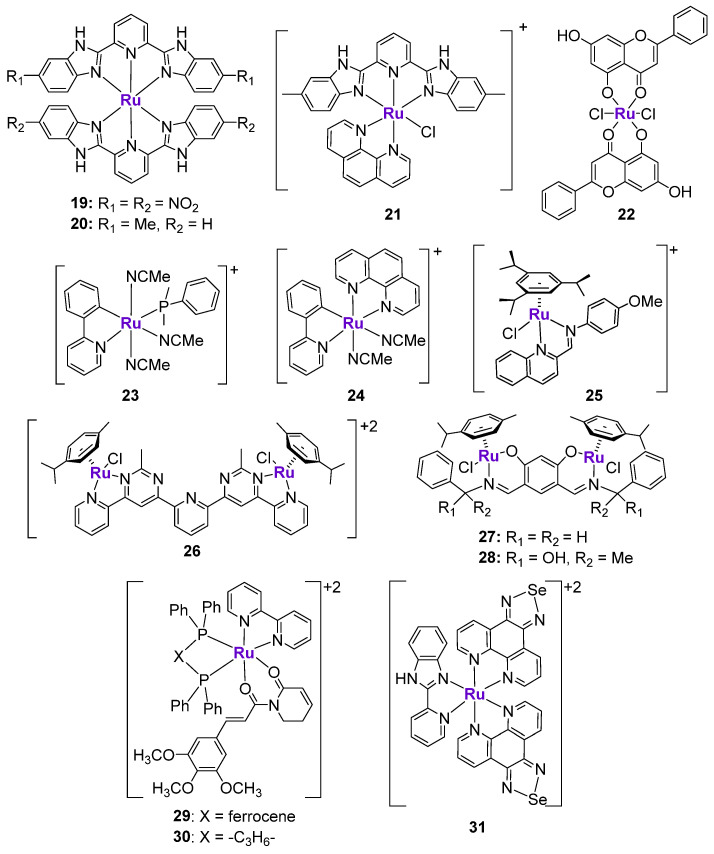
Ru complexes affecting p53 expression.

**Figure 6 cancers-15-02834-f006:**
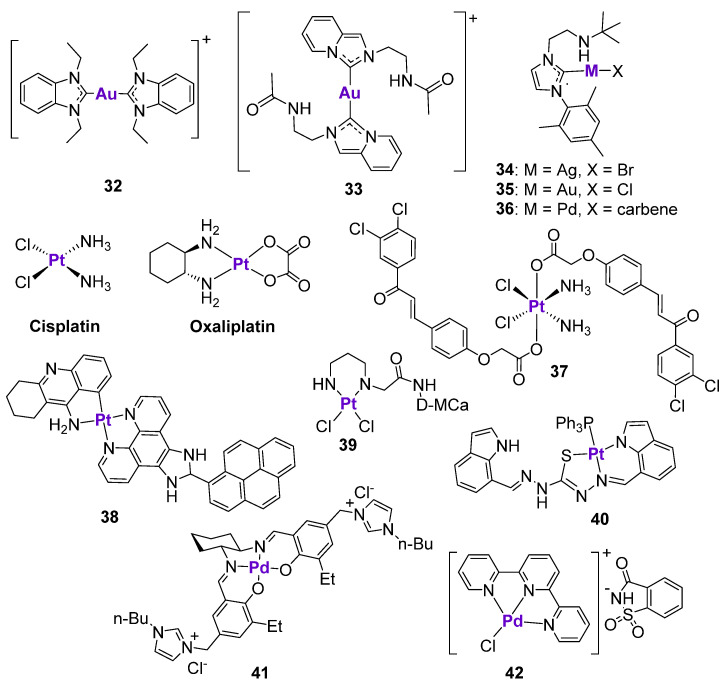
Ag, Au, Pt and Pd complexes affecting p53 expression.

**Figure 7 cancers-15-02834-f007:**
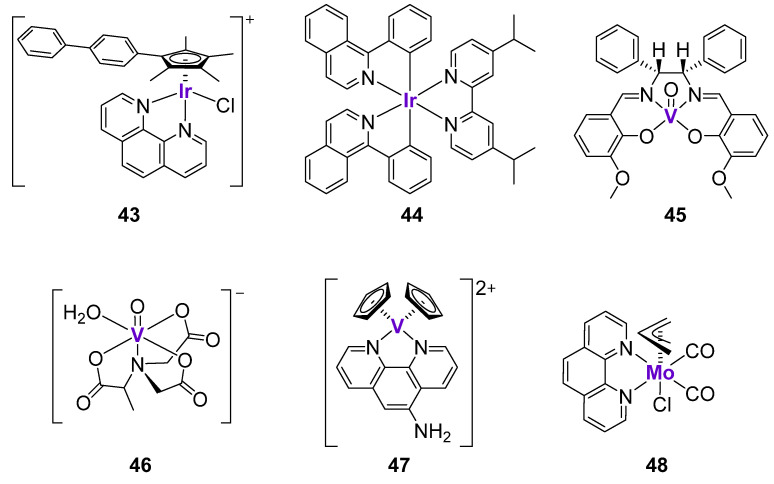
Cytotoxic Ir and V complexes affecting p53 expression.

**Figure 8 cancers-15-02834-f008:**
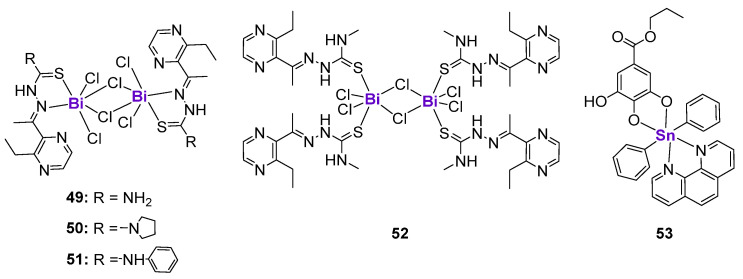
Bi and Sn complexes affecting p53 expression.

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
