# Peer review of "Metal-Based Anticancer Complexes and p53: How Much Do We Know?"

_cancers, 2023, doi:10.3390/cancers15102834_

Round 1

Reviewer 1 Report

This is a very timely review to revisit the role of anticancer metal complexes against p53 as a target. The manuscript is well-written and clearly structured. It highlights the key factors when considering metal complexes as p53 targeting drugs and provides an exhaustive and summative discussion of the major developments in the field. My comments are as follows:

1) p53 has been long considered as an undruggable target since wildtype p53 is easily degraded, while mutant p53 become stabilized and accumulate in cancer cells. Is there any studies that investigates whether metallodrugs can selectively promote mutant p53 degradation considering that most of current studies focus on translational regulation of p53?

2) The authors have done comprehensive literature review and extracted key information from literature. It would be highly useful if the authors can share more their insightful perspectives in the field in the future. For example, where could the development of p53-targteing metal complexes go moving forward? and what are some of the remaining challenges/big questions in this field?

This is an excellent review, in terms of breath and depth, which would be of high interest to readers of this journal. I recommend publication in its current form, following clarification of minor comments.

Author Response

>> We thank the reviewer for critical evaluation of our work and valuable suggestions. Please find our answers below.

  • p53 has been long considered as an undruggable target since wildtype p53 is easily degraded, while mutant p53 become stabilized and accumulate in cancer cells. Is there any studies that investigates whether metallodrugs can selectively promote mutant p53 degradation considering that most of current studies focus on translational regulation of p53?

>> This is an excellent suggestion. We found several compounds that were able to induce the degradation of mutant p53 and added the description in the text.

“Subsequently, the comparison of the p53 levels in SKBR3 cells (R175H mutation) and HCT116 (wild-type p53) treated with equimolar concentrations of complex 1 revealed the time-dependent decrease of p53 levels in SKBR3 cells and time-dependent increase of p53 levels in HCT116 cells, without affecting p53 mRNA levels [70]. It was revealed that reactivation of mutant p53 protein by complex 1 triggered the process of autophagy, leading to the degradation of remaining mutant p53 proteins. In agreement, inhibition of autophagy by specific autophagy inhibitors or inhibition of p53 transactivation by pifithrin-a abrogated the degradation of mutant p53 proteins [70].
Additionally, the inhibition of ER stress impaired the autophagy induced by complex 1, as well as the degradation of mutant p53 protein
[71]. Therefore, it was concluded that the prevalence of wild type over mutant p53 function was dependent on the ability of complex 1 to shift the balance between folded and misfolded p53 proteins and trigger UPR activation [71].”

“Besides small Zn-containing molecules, several groups investigated the effects of Zn-containing nanoparticles. For example, photothermal Zn-doped Prussian blue nanoparticles were designed to release Zn2+ ions inside cancer cells, thereby triggering apoptosis and autophagic degradation of a mutant p53 protein [73]. Similar effects were induced by Zn-Fe nanoparticles, which induced degradation of p53 proteins with different mutations, but not the degradation of wild-type p53 [74].” 

“Interestingly, arsenic trioxide (As2O3), which is used for the treatment of acute promyelocytic leukemia (APL), demonstrated very pronounced effects on the stability of mutant p53 proteins [165]. Treatment of cancer cells with wild-type p53, such as MCF-7, RKO, HCT116 and MEF, with arsenic trioxide resulted in the dose-dependent increase of p53 expression. On the contrary, HaCaT, SW480 and MIA PaCa-2, were characterized by the time- and dose-dependent reduction of p53 expression. This drug induced proteosomal degradation of mutant p53 proteins with various mutations, including p53R175H, p53H179Y/R282W, p53R248W and p53R273H. Additionally, As2O3 decreased the stability of mutant p53 proteins by blocking its shuttling between nucleus and cytoplasm [165].”

  • The authors have done comprehensive literature review and extracted key information from literature. It would be highly useful if the authors can share more their insightful perspectives in the field in the future. For example, where could the development of p53-targteing metal complexes go moving forward? and what are some of the remaining challenges/big questions in this field?

>> Following reviewer’s suggestion, we extended the conclusions and added the remaining challenges in the discussion.

“What do we really know about p53-targeting metal complexes? The analysis of the literature revealed that this field remains largely unexplored. With the exception of Zn complexes, the majority of scientific studies on p53-dependent anticancer metal complexes did not include the information on their interactions with a p53 protein and only demonstrated the effects of metal complexes on p53 expression of cancer cells. One should keep in mind that due to the critical role of p53 in the function of normal cells, the effects of p53-targeting metal complexes might be potentially toxic. For example, the toxicity induced by the clinically used drug cisplatin was linked to its activation of p53; however, it could be prevented or reversed by the co-administration of p53 inhibitors. Hence, it would be interesting to study whether the p53-targeting metal complexes can induce p53 activation in normal tissues and whether it can be reversed by the addition of p53 inhibitors, such as pifithrin-a. We believe that more in-depth mechanistic studies on metal complexes in the p53 context, e.g. effects on the protein conformation and degradation, would be very desirable and could stimulate the preclinical development of metal-based drug candidates.”

Reviewer 2 Report

The review of Babak et al. describes the role of several endogenous metals in the function and structure of p53 protein, as well as the design of metal complexes with certain effects on p53 and its family members. I think this review will be interesting for the readership of Cancers journal after several minor revisions.  I like that each section has several sentences describing the importance of metal complexes in cancer, but I didn’t find such an introduction for copper complexes. The authors should describe the state of the art of Cu complexes in cancer and add several important references, such as DOI: 10.1126/science.abf0529 and 10.1002/cbic.202300033.  Also, I am not sure why the authors combined Au, Ag, and Pd in one section. It would be more logical to split this section into two sections: 1) Au and Ag and 2) Pt and Pd.        

Author Response

The review of Babak et al. describes the role of several endogenous metals in the function and structure of p53 protein, as well as the design of metal complexes with certain effects on p53 and its family members. I think this review will be interesting for the readership of Cancers journal after several minor revisions.

>> We thank the reviewer for critical evaluation of our work and valuable suggestions. Please find our answers below.

I like that each section has several sentences describing the importance of metal complexes in cancer, but I didn’t find such an introduction for copper complexes. The authors should describe the state of the art of Cu complexes in cancer and add several important references, such as DOI: 10.1126/science.abf0529 and 10.1002/cbic.202300033. 

>> Following the reviewer’s suggestion, we added several introductory sentences about Cu complexes in cancer, as well as requested references.

“Various structurally different Cu complexes were shown to be highly cytotoxic in different cancer cell lines [73,75]. Since intracellular Cu balance is tightly regulated, the overload of bioavailable Cu leads to the fatal Cu overload and cancer cell death [73]. The Cu-induced cell death mechanism was recently termed cuproptosis [76]. It was suggested that cuproptosis might be regulated by p53 [77]. Besides cuproptosis, the mechanism of action of Cu complexes is typically linked to the induction of ROS based on the Fenton-like mechanisms [78]. There are several Cu complexes, whose anticancer effects were investigated in the p53 context.”  

Also, I am not sure why the authors combined Au, Ag, and Pd in one section. It would be more logical to split this section into two sections: 1) Au and Ag and 2) Pt and Pd.  

>> Following the reviewer’s suggestions, we split the sections into 1) Ag and Au and 2) Pt and Pd. In addition, we significantly extended the section on Pt and Pd and added information about clinically used Pt drugs, as well as several experimental Pt-based complexes.

Reviewer 3 Report

The manuscript by Alfadul et al summarized the current knowledge on the interaction of metal complexes with P53. Considering that metal complexes are frequently supposed to function via induction of DNA damage, it is rather surprising that this topic has not been reviewed so far. Consequently, the review is very timely and elegantly covers the different aspects how metal drugs can interact with P53 (not only by DNA damage-induced activation). The review is well written and nice to read. I have only two minor comments which should be addressed before publication:

- chapter 2.1.3. I would suggest to include here also the literature on COTI-2, because this is a clinically investigated TSC derivative, which is also reported to restore some P53 mutations.

- chapter 2.3.1  Maybe it is worth mentioning here the data on KP772 (a lanthanum complex similar to compound 15), for which it has been shown that P53 activation is not involved in its anticancer activity.

Author Response

The manuscript by Alfadul et al summarized the current knowledge on the interaction of metal complexes with P53. Considering that metal complexes are frequently supposed to function via induction of DNA damage, it is rather surprising that this topic has not been reviewed so far. Consequently, the review is very timely and elegantly covers the different aspects how metal drugs can interact with P53 (not only by DNA damage-induced activation). The review is well written and nice to read. I have only two minor comments which should be addressed before publication:

>> We thank the reviewer for critical evaluation of our work and valuable suggestions. Please find our answers below.

- chapter 2.1.3. I would suggest to include here also the literature on COTI-2, because this is a clinically investigated TSC derivative, which is also reported to restore some P53 mutations.

>> We included COTI-2 in the discussion in section 1:

“Although there are many small molecules that are currently being tested in clinical trials or at the advanced preclinical stage (e.g. mutant p53 activator COTI-2) [34,35], none of the molecules targeting p53 and other members of p53 family has been clinically approved yet [35].”

- chapter 2.3.1 Maybe it is worth mentioning here the data on KP772 (a lanthanum complex similar to compound 15), for which it has been shown that P53 activation is not involved in its anticancer activity.

>> That’s a great suggestion! Totally makes sense. We added the following discussion in section 2.3.1:

“These results possibly indicate that the mechanism of action of more lipophilic complexes 15-18 might be both p53-dependent and p53-independent. The role of a metal center on p53 activation might be assessed by the comparison of tris(1,10-phenanthroline)Fe(III) complex 15 and its structural La(III) analogue (KP772) [85]. KP772 induced expression of p53 and p21Waf1 in A549 cells with a wild-type p53 status. However, subsequent comparison of the effects of KP772 in p53-null Hep3B-cells and its p53-transfected analogue (Hep3B/p53) revealed no significant differences in its activity, suggesting that, in contrast to Fe(III) complex 15, KP772 might act largely through p53-independent mechanisms [85].”